# Life Experience Matters: Enrichment and Stress Can Influence the Likelihood of Developing Alzheimer’s Disease via Gut Microbiome

**DOI:** 10.3390/biomedicines11071884

**Published:** 2023-07-03

**Authors:** Sarah E. Torraville, Cassandra M. Flynn, Tori L. Kendall, Qi Yuan

**Affiliations:** Biomedical Sciences, Faculty of Medicine, Memorial University of Newfoundland, St. John’s, NL A1B 3V6, Canada

**Keywords:** Alzheimer’s disease, environmental factors, stress, enrichment, gut microbiome, chronic mild stress, gut dysbiosis, probiotics

## Abstract

Alzheimer’s disease (AD) is a chronic neurodegenerative disease, characterized by the presence of β-amyloid (Aβ) plaques and neurofibrillary tangles (NFTs) formed from abnormally phosphorylated tau proteins (ptau). To date, there is no cure for AD. Earlier therapeutic efforts have focused on the clinical stages of AD. Despite paramount efforts and costs, pharmaceutical interventions including antibody therapies targeting Aβ have largely failed. This highlights the need to alternate treatment strategies and a shift of focus to early pre-clinical stages. Approximately 25–40% of AD cases can be attributed to environmental factors including chronic stress. Gut dysbiosis has been associated with stress and the pathogenesis of AD and can increase both Aβ and NFTs in animal models of the disease. Both stress and enrichment have been shown to alter AD progression and gut health. Targeting stress-induced gut dysbiosis through probiotic supplementation could provide a promising intervention to delay disease progression. In this review, we discuss the effects of stress, enrichment, and gut dysbiosis in AD models and the promising evidence from probiotic intervention studies.

## 1. Alzheimer’s Disease: Still No Cure

Dementia is a set of symptoms, including but not limited to memory loss, problem-solving issues, and mood changes, spanning decades and not considered a “normal” part of aging [1,2]. Alzheimer’s disease (AD) is the leading cause of dementia, being responsible for 60–80% of dementia diagnoses and typically, though not exclusively, affecting people aged 65 years and older. AD is a chronic neurodegenerative disease, characterized by the presence of β-amyloid (Aβ) plaques and neurofibrillary tangles (NFTs) formed from abnormally phosphorylated tau proteins (ptau) [3]. Aβ plaques accumulate in the extracellular space between neurons, while NFTs accumulate within the neurons themselves, each preventing normal cellular functioning and communication, eventually leading to their deterioration [4,5]. Aβ and NFT can be examined post-mortem, as well as detected using fluid and imaging biomarkers in live patients [3,6]. Symptoms of AD appear to gradually and progressively worsen from mild cognitive impairment (MCI) to severe impairment, changes in personality, and physical disabilities requiring constant care.

Drug treatment options have focused primarily on mid-to-late-stage AD and have historically targeted the cholinergic system implicated in memory decline or the Aβ cascade [7,8]. These drug treatments have been largely ineffective, demonstrating, at most, some improvement in cognition but still an overall decline as time progresses. In addition to this, new treatments are being explored targeting tau and tau mechanisms. Drugs targeting post-translational modifications of tau such as inhibition of tau hyperphosphorylation kinases (GSK3B, CDK5, Fyn), promoting the activity of tau dephosphorylation, or inhibiting tau acetylation are in phase 1 and 2 of various clinical trials [9,10,11]. These drugs include rapamune, nilotinib, lithium chloride, salsalate and MK8716. Drugs targeting tau aggregation (methylene blue, curcumin), microtubule stabilization (NAP, TPI 287), tau expression and tau clearance through immunotherapy (C2N-8E12, BMS-986168, AADvac-1) are currently undergoing trial [12]. Despite promising preclinical results, many trials targeting tau have failed due to adverse effects [13]. Methylene blue, which showed great promise in animal studies, was shown to be limited to inhibition of tau fibril formation and does not provide the same effect to tau oligomers [10]. Other therapeutic strategies such as anti-inflammation have been reviewed elsewhere [13]. From these drug trials, it is becoming evident that pathological processes begin long before clinical symptoms, and no significant clinical benefits have been observed. Given the gradual and progressive nature of AD, an emphasis must be put on finding targets for treatment that are present during the earlier, pre-clinical stages of AD progression. This window for treatment spans years to decades, therefore providing ample opportunity to change potential outcomes.

## 2. Stress, Enrichment and the Gut Microbiome: Potential Focus for a Cure

While there is evidence of AD being highly heritable, ranging from approximately 60–100% heritability depending on the timing of disease onset, the prevalence of genetically linked AD cases is very low (<1%) amongst all cases of the disease [14,15,16,17]. This suggests that there are other non-genetic, environmental risk factors that play a larger role in determining whether an individual develops AD and, therefore, may provide avenues for alternative treatment options.

### 2.1. Stress and AD

Up to 40% of Alzheimer’s risk can be attributed to environmental factors [18]. Disease development and progression can be strongly influenced by stress, although the presence of stress alone may not lead to Alzheimer’s disease. Stress may be one of the many factors that influence whether symptoms will appear earlier or later when a precursor to the disease (i.e., persistently phosphorylated tau) is present. Stress-induced physiological processes can adversely affect healing, coping, and maintaining a positive quality of life. Cognitive and motor functions are deteriorated by neurodegenerative diseases, which is by itself a form of stress. Furthermore, AD also leads to dysregulation of the stress response through the HPA axis [19].

Stress is well-known to play a negative role in the development of a variety of disorders such as cardiovascular disease, obesity, and gastrointestinal disorders, as well as psychiatric and neurodegenerative disorders. According to work on animal models, manipulating the glucocorticoid milieu can result in behavioural, molecular, and cellular changes similar to AD [20,21,22]. Specifically, stress or glucocorticoid administration increases Aβ precursor protein and tau phosphorylation, both of which are associated with synaptic dysfunction and neuronal death in AD [21,22].

Several studies have found direct links between stress and Alzheimer’s disease pathology. A study published by Sotiropoulos and colleagues found that stress resulted in abnormal phosphorylation of tau [18]. Stress upregulates at least two tau epitopes, which are strongly implicated in the pathophysiology of Alzheimer’s disease. In another study on more than thirteen thousand patients tracked over the course of fifty years, a two-fold greater risk of dementia in the late stages of life in patients who suffer from depression was observed [23]. An analysis of study results from the Rush Memory and Aging Project showed that patients who demonstrated a high level of “distress proneness” were three times more likely to develop dementia over the course of three years [23]. Neuroticism, which refers to the tendency to experience distress and negative emotions, has been linked to a wide variety of mental health conditions. It is known that individuals with high levels of neuroticism are more likely to suffer from depression, anxiety disorders, and post-traumatic stress disorder [24]. Also, evidence suggests a link between neuroticism and Alzheimer’s disease. An 800-member cohort of the Catholic clergy was followed for five years by Wilson and colleagues, in which participants were assessed for distress proneness at baseline [25]. They found that higher levels of neuroticism at baseline were associated with a higher risk of developing AD dementia over time, namely, people who were distressed had a 2.4 times higher risk of developing dementia [25].

It has been found that patients with AD possess increased cortisol levels in their biological fluids (e.g., plasma, saliva, and cerebrospinal fluid) [20], which correlates with the high association between anxiety-related mental illness and AD [26]. Researchers have found that those with mild cognitive impairment typically have higher average circulating cortisol levels than their age-matched counterparts at all diurnal time points [24]. Moreover, patients with dementia exhibit impaired hypothalamic–pituitary–adrenal (HPA) axis feedback when dexamethasone is given, indicating impaired HPA axis function. The release of cortisol is expected to increase with increased HPA activity, accelerating and intensifying disease progression.

### 2.2. Enrichment Effects on Cognition and AD

Environmental enrichment can also influence the development and progression of dementia. Enrichment is often examined as cognitive (i.e., mentally stimulating or requiring mental processing) or physical activities. Engaging in cognitively stimulating activities may alter the AD pathophysiological cascade in a positive manner among older adults. In a longitudinal study examining life-long cognitive enrichment, including education and other mentally stimulating activities, it was found that high cognitive enrichment scores throughout the lifespan reduced the risk of developing cognitive impairment, and delayed progression from mild cognitive impairment to dementia [27]. Among cognitively normal older adults, Wilson et al. [28] found that following a 6-year long, multi-interview assessment of frequent cognitive activity, with activities such as reading, watching television or listening to the radio, playing games or completing puzzles, and going to museums, each additional point of cognitive activity reduced cognitive decline by 52% during follow up assessments. Leisure activities that stimulate cognitive function have been associated with a reduced risk of dementia development. For example, crossword puzzles may reduce the risk of cognitive decline by improving cognitive reserve, by directly modifying the disease, or by integrating a variety of other healthy behaviours [29]. It has been reported that individuals engaged in cognitively stimulating leisure activities experienced slower rates of cognitive decline and a faster rate of post-onset cognitive decline—this is consistent with the cognitive reserve hypothesis, which holds that cognitively stimulating activities may delay the development of clinical cognitive impairments [30,31].

The results of psychological studies have suggested that conversation is a highly cognitively stimulating task. To understand others’ intentions and feelings, in addition to linguistic ability, conversations require attention, working memory, organizational and control of thoughts (executive functions), and social cognition. Across decades of research, isolation, which older adults are particularly susceptible to, has consistently been linked to cognitive decline and AD. Compared with their counterparts who had five or six social connections, elderly individuals with no social connections were 2.37 times more likely to experience cognitive decline in one cohort study [32]. Strengthening this, loneliness increases the risk of AD by more than double [33] and has been found to spike levels of Interleukin-6, an inflammatory agent implicated in numerous age-related diseases, including AD [34].

One enriching activity is dancing, which has grown in popularity amongst seniors in recent years as it integrates a variety of aspects, from audiovisual perception to physical perception, and emotional expression. This method of non-pharmacological cognitive intervention is beneficial to older adults because it improves episodic memory, executive performance, and global cognition [35]. Additionally, the use of music, which engages the entire brain, including the auditory system, syntactic system, semantic system, memory, and motor function, can contribute to the enhancement of cognitive function.

Studies have demonstrated that mild to moderate levels of physical activity can reduce the likelihood of dementia and AD. In a systematic review of 163,000 non-psychotic participants, researchers discovered that physical activity could reduce the risk of dementia by 28% and AD by 45% [36]. Various studies have linked exercise with hippocampus development as it reduces the decline of cortical tissue in the elderly. MRI findings showed that active individuals had a larger hippocampus, better spatial memory, and improved cognitive and physical health. There is a growing body of evidence that exercise reduces cognitive degeneration and dementia [37].

### 2.3. Microbiota–Gut–Brain Axis and AD

The microbiota–gut–brain axis represents the bidirectional communication between the gut and the brain. In AD, gut dysbiosis is becoming increasingly important due to the effects it can have on gut barrier function, blood–brain barrier (BBB) function, and neuroinflammation. Increased barrier permeability, as seen in AD studies, provides an opportunity for toxic amyloids, neuroinflammatory cytokines, lipopolysaccharides (LPS), and short-chain fatty acids (SCFAs) to move into the brain and seed neurodegeneration [38]. The microbial community of the gut microbiota plays an active role in homeostasis and disease. In humans, aging contributes to the increase in pro-inflammatory bacteria, which has been found to induce systemic inflammation. Pro-inflammatory bacteria and, in turn, neuroinflammation has been found in amyloid-positive patients when compared to healthy elders [39]. Human studies aiming to classify differences in bacterial species in the gut makeup of healthy vs. AD patients have provided insight into the interplay between human health and disease. Vogt et al. discovered a significant decrease in *Firmicutes* and *Bifidobacteria* in the fecal samples of AD patients, accompanied by an increase in *Bacteroidetes* species [40]. This was validated through 16sRNA sequencing conducted by Zhuang et al. and Liu et al., finding similar results with the addition of the dysregulation of multiple other bacterial families [41,42]. Interestingly, a difference between Enterobacteriaceae was also found between MCI and AD patients, indicating a change in gut composition with disease progression [42]. A further understanding of the regulation of bacterial species through aging and disease is needed, as vast discrepancies can be found between patients, demographic regions, and ages.

### 2.4. Stress and Gut Dysbiosis

The complex system of the gut microbiota is constantly challenged by biological and lifestyle factors, including environmental factors like stress. Studies on military personnel reveal the detrimental effects of chronic stress in extreme situations and often multiple stressors in combination [43]. In this example of extreme chronic stress, many health decrements have been observed including musculoskeletal injury [44], endocrine disruption [45], inflammation [46], and illness and infection [47]. Military personnel are a common group to study changes from extreme stress in gut microbiota, however, various levels of stress are a ubiquitous part of daily life for many individuals.

Commonly, stress induces the release of hormones such as glucocorticoids and catecholamines which have been shown to modulate GI function and microbial growth [48,49,50]. Stress-induced changes along the vagus nerve have also been found to reduce digestive activity, likely altering substrate availability [50]. It has been shown that stress-induced changes in gut microbiota in humans can degrade the physical gut barrier and increase gut permeability [51].

In a longitudinal study on military personnel, 73 soldiers were placed in a situation of high stress and examined for microbiota composition changes and intestinal permeability [52]. It was found that 23% of metabolites were significantly altered after stress, and increased intestinal permeability was observed through an increase in sucralose excretion during stress, independent of diet [52]. Additional longitudinal studies were performed on psychological stress in students before and during exams [53,54]. In students, it was found that fecal levels of lactic acid bacteria were lower during a high-stress situation, associated with an increase in cortisol concentration [53]. Strengthening this, higher levels of salivary cortisol were observed before an exam [54]. The daily consumption of probiotic *Lactobacillus casei* significantly reduced gastrointestinal stress and symptoms and was found to preserve the diversity of the gut microbiota compared to non-probiotic-fed students [54].

## 3. Animal Models of Stress, Enrichment, and Gut Dysbiosis in AD

As aforementioned, environmental factors are implicated in the development of symptoms related to dementia and AD, including environmental enrichment, stress, and gut microbiome health [55,56,57]. Environmental enrichment, including enhanced living conditions, increased social encounters, and a variety of activities has been shown to improve cognitive abilities and reduced symptoms of stress and depression often comorbid with AD. Conversely, stress, particularly chronic stress, is one of the largest risk factors associated with AD and has been implicated in both the onset of the disease and in worsening disease progression. The bi-directional communication between the gut and the brain is also thought to play an important role in disease progression on its own and can be altered by both stress and enrichment. The effects of enrichment, stress, and gut health on AD are frequently studied in animal models and those models will be discussed below.

### 3.1. Roles of Enrichment in Animal Models of Alzheimer’s Disease

A large number of animal models of environmental (EE) and social (SE) mimicking cognitive enrichment in humans, along with physical (PE) enrichment, have shown that enrichment largely improves cognitive symptoms of AD and ameliorates underlying pathological processes and inflammatory responses and reduces neuronal death [58,59,60,61,62,63,64,65]. Transgenic mouse and Aβ-infused rat models of AD using PE, including voluntary running exercise and resistance ladder climbing, showed improvements in both short- and long-term memory, reduced inflammation, and reversed deficits in neurogenesis compared to non-physically enriched AD counterparts [60,61,65,66,67]. However, PE is not as effective as other forms of enrichment alone and is often used in combination with or as part of other enrichment paradigms. Similarly, SE, achieved through grouping animals together either in housing or for the duration of the paradigm, in AD models also shows promising results, with animals having intact social recognition, improved spatial learning, improvements in motor skills, and reduced anxiety-like symptoms compared to non-enriched transgenic or Aβ-infused AD model animals [60,62,63,68], like with PE, SE alone does not produce as strong of effects as more comprehensive or combined protocols of enrichment. EE typically involves providing subjects with a variety of items that can be interacted with, including toys, chews, hides, treats, and running wheels. In AD models, EE has been shown to improve recognition and spatial memory, reduce tau and Aβ pathology, improve neurogenesis, reverse effects of stress and high-fat or high-sucrose diets, reduce inflammation, increase synaptic plasticity, improve immune system activity, and reduce neuronal death seen in AD [58,59,60,61,62,63,66,68,69,70,71,72,73,74,75,76,77,78,79,80,81,82,83,84,85,86,87]. These vast beneficial effects of EE are seen in both mouse and rat models, with enrichment protocols occurring over timelines beginning between 3 weeks of age to 18 months, and lasting between 1 month and 23 months. Results across varying timelines are relatively consistent, though some studies have found time-dependent effects of enrichment on AD pathology and symptom development. For example, Mirochnic and colleagues found that 6 months of enrichment was sufficient for EE, but not PE, to elicit beneficial effects, but both were beneficial after 18 months [66]. Similarly, Polito and collaborators [88], found that following 15 months of EE, Aβ deposition was reduced; however, 4 months was not long enough to produce the same effect despite being sufficient to reduce the spatial memory deficit seen in AD. Others have implemented EE protocols prior to AD pathology appearing and found that it can have protective or preventative effects [60,74,76,89].

Another strategy to examine the protective effects of EE has been to implement enrichment protocols during gestation or neonatal stages. Ziegler-Waldkirch and colleagues [90] found that pregnancy and lactation increased Aβ pathology in an APP/PS1 transgenic mouse model, but exposure to EE during pregnancy and the lactation period prevented such pathology, with those dams having as little Aβ and as much neurogenesis as non-pregnant counterparts. Furthermore, another study found that EE during pregnancy of AD transgenic left offspring with intact learning and memory and synaptic plasticity, suggesting generational protective effects [91]. Gentle handling during the neonatal period has produced some mixed results. One study found that this form of enrichment early in life was able to reduce behavioural traits of AD (i.e., increased swim time and reduced immobility time in a forced swim test) more effectively than EE later in life [92]. On the other hand, a study found neonate handling to only produce mild protective effects against spatial memory decline, though that effect was more pronounced in females than males [93].

There are also examples of rather unique enrichment protocols. One study, conducted by Yeung and colleagues [64], used repeated cognitive testing as intellectual or cognitive enrichment in their 3xTG-AD mouse model of AD and found that this regular training improved performance on a separate spatial test compared to non-enriched AD animals. Another study used olfactory enrichment, in which a different odor was presented in the animals’ home cage each day over 40 or 80 days [94]. They found that long-term olfactory enrichment reduced tau phosphorylation. Results from these studies demonstrate that a variety of activities and stimuli can be considered enriching and produce beneficial effects to protect against AD.

Enrichment does not always produce such promising results, however. A number of studies have shown that EE produces rather mild or mixed results, especially during the later stages of AD pathology. These findings include failing to reduce Aβ or tau pathology, cognitive decline, or neuronal loss, and not improving cognitive performance, inflammation levels, or neurogenesis [95,96,97,98,99,100,101,102,103,104,105,106]. These works have not necessarily been unsuccessful in demonstrating some improvements following EE, but the effects are not as robust as the studies discussed above. Additionally, some studies have found that EE worsens AD pathology, resulting in reduced neurogenesis, increased apoptosis, and more rapid and extensive accumulation of Aβ despite cognitive improvements in AD animals, while producing opposite results in healthy counterparts [107,108,109,110]. These studies suggest that enrichment may facilitate some pathological processes of AD due to genetic predisposition. Fulopova and collaborators [111], also provided a possible explanation for the varied results of enrichment studies by demonstrating that Aβ accumulates more so in the prefrontal cortex than in other brain regions, and while EE is able to improve cognitive function and reduce Aβ accumulation, the prefrontal cortex maintains high levels of plaques.

Some experimenters have recognized that EE alone is not sufficient to produce the desired widespread improvement in cognitive function and pathology in AD and have used other therapies alongside enrichment to assess their combined effect. Studies combining EE with drug therapies, such as memantine, an N-methyl-D-aspartate receptor antagonist, donepezil, an acetylcholinesterase inhibitor, magnesium-L-threonate, an ingestible method to increase essential magnesium in the brain, or *Bifidobacterium breve* probiotic, have shown that the drug or EE alone can produce some benefits, but when combined there is improvement in learning and memory or more long-term effects [112,113,114,115]. Combination therapy using EE and caffeine was not as robust, with individual therapies having nearly equal effects as combined therapies [116]. Alternatively, one study using an aluminum induction mouse model of AD combined EE with resveratrol treatment, an antioxidant and anti-inflammatory compound, and found that while both individually reduced Aβ deposition, when combined the Aβ burden worsened [117].

Ultimately, these enrichment studies provide a hopeful message that a variety of enriching stimuli, including enjoyable activities, a varied diet, physical activity, social interactions, mentally stimulating activities, and even olfactory stimulation, can reduce the risk and severity of AD pathology and symptom development. Enrichment may counteract some negative environmental influences, including stress and an unhealthy diet, that would worsen the risk of AD, and if not sufficient alone, can be used in combination with other therapies to combat symptom development in AD. These enrichment AD models are summarized in Table 1 below.

### 3.2. Roles of Stress in Animal Models of Alzheimer’s Disease

Animal models examining the roles of stress, in its many forms, and stress-associated hormones in AD have demonstrated a link between environmental stress and pathological progression and symptom development of AD, often concluding that corticotrophin-releasing factor (CRF) is an essential mechanistic factor in the said link [18,62,119,120,121,122,123,124,125,126,127,128]. Further, this relationship is bidirectional, with stress acting as a driving force behind AD-like symptoms and pathology development, shown from studies finding AD pathology and symptoms in wild-type (WT) subjects post stress, but also making subjects more prone to stress and susceptible to its effects [129,130,131,132,133,134,135].

One common stress paradigm used in animal models of AD is chronic unpredictable stress (CUS), typically involving a number of different stressors applied in random—or unpredictable—order over three to six weeks [18,62,122,129,133,134,136,137,138,139,140,141]. Frequently used stressors include soiled bedding, a tilted cage, bright lights or disrupted light schedule, and changing litter mates. Studies using such methodology have shown that this leads to increased Aβ deposition, increased ptau, impaired spatial memory, reduced synaptic plasticity, increased impulsivity, and impaired olfactory discrimination learning.

Similarly, sleep deprivation (SD) is another prominent method used to elicit stress in AD models and is fairly effective at inducing AD pathophysiology, with SD in WT animals usually resulting in reduced cognitive function and accumulations of Aβ and ptau [130,131,142]. There is great variability in the duration of protocols of SD, with some as short as a single 4 h session, and others lasting up to 6 weeks [130,143]. The most common method to achieve SD in rodent models of AD is using a modified cage filled with water and only small platforms throughout so that animals are still able to move around but will fall into the water when they fall asleep, which results in complete SD or fragmented sleep. Lack of sleep has devastating effects on spatial memory, fear learning, and synaptic plasticity, and results in increased Aβ and ptau deposits [143,144,145,146,147,148,149,150,151,152,153,154]. Difficulty sleeping is also one of the ways in which the bidirectional link between stress and AD is demonstrated. As described, SD or fragmentation has been shown to increase AD pathology; however, Kincheski and collaborators [144] infused Aβ into the brains of subjects and found them to have more fragmented sleep. This two-way relationship between stress and AD can create a cycle, with stress triggering AD pathology, which, in turn, increases levels of stress.

Other common stress paradigms used include restraint and immobility (RIS) stress and social isolation (SI). Like with CUS and SD, RIS and SI also increased levels of Aβ and ptau, with one study finding this to be specific to female subjects after 5 days, poorer spatial and fear memory, increased inflammation, increased neurodegeneration, and CRF implicated as a mediating factor for these changes [120,121,124,155,156,157,158,159,160,161,162]. Despite this, some studies have found that RIS alone may not be sufficient to elicit strong effects, either with behavioural or pathological changes, suggesting that some stressors may not be as detrimental to health as others [163,164].

Studies examining the hormonal aspect of stress (HS) and its relationship to AD have concluded that glucocorticoids can lead to increased soluble Aβ and impair fear recall, while CRF and its receptor, corticotrophin-releasing factor receptor-1, are involved in the process of ptau phosphorylation [115,119,123,128,151]. Other studies have examined alternate styles of stressors, including maternal separation (MS) and psychosocial stress (PS), and found that these stressors lead to worsened spatial memory, increased levels of stress hormones, increased inflammation, and increased Aβ and ptau in AD animals [125,126,165,166,167].

These animal studies create a clear image that stress is linked to the pathological progression of AD and symptom development. Stressors have been applied at time points from during pregnancy, early in life, and into late adulthood, and largely have resulted in worsened cognition and increased AD pathology. These models, along with the models of enrichment discussed earlier, create a strong case encouraging people to avoid stress and live an enriched life. A summary of the stress models is provided in Table 2 below.

### 3.3. Stress and Gut Microbiota in Animal Models

The gut microbiota has been identified to play a key role in regulating brain function. The bidirectional network between the gut and brain is connected by various pathways including the vagus nerve, neurotransmitters, immune system, hypothalamic–pituitary–adrenal (HPA) axis, and metabolites [67,168,169]. The modification of the gut microbiota and gut community makeup has gained attention in recent years and positive outcomes have been shown in various diseases.

The gut undergoes periods of shift, where significant changes and regulation of bacterial makeup occur at specific time points in one’s life [170]. Interestingly, these periods can be targeted through stress paradigms, and the important gut establishment can be disrupted. Specific changes in gut microbiota communities due to stress-inducing paradigms are outlined below.

#### 3.3.1. Gut Dysbiosis and Stress

Stress has been shown to have varying effects on gut dysbiosis and gut health at all stages of life. Studies in germ-free (GF) mice have provided evidence for clear communication between the gut and the brain, termed the “microbiota gut–brain axis” [171]. It was found that in GF mice, mild restraint stress produced an upregulation of corticosterone and adrenocorticotropic hormone, indicating a critical role of the microbiota in the stress response and HPA axis [171]. Recent research has provided strong evidence to show that gut microbiota makeup is related to stress in both animal models and humans. Further strengthening the relationship between stress and the gut, stressors presented in animal studies have been shown to negatively affect gut health and cause gut dysbiosis (Table 3). Together, these studies present evidence that stress can be both a cause and a result of gut dysbiosis.

It has been well documented that stress exposure can have a direct influence on the gut microbiota makeup, and can damage the balance between microbes. A plethora of bacteroids has been found to be regulated through chronic stress paradigms in animal models. Specifically, maternal separation (MS) is used as a common model of stress. MS is based on the work of Hofer, where the deleterious impact of early weaning was revealed, and shown to have implications on offspring physiology and intestinal health [172]. The model uses the early mother–infant separation in animals that most commonly occur during the first 1–3 weeks of life [173]. This development period is a critical window of both sensitivity to stress and microbiota community establishment [174,175]. A growing body of evidence suggests MS paradigms can alter the gut microbiota composition in both animal and human models. However, the quantification and direction of gut disturbance are not consistent across studies due to the use of different species, strains, sex, protocols, and microbiota analyses.

Levels of fecal and intestinal bacteria have been shown to be altered in many animal studies of MS. As shown in Table 3, studies in mice induced gut microbiota alterations between stressed animals and control groups. Specifically, the abundance of bacteria and diversity levels in the microbiome were negatively affected by the stress paradigm. In support of this, in one study on postnatal day (PD)-28, mice showed a reduced level of alpha diversity in the gut [176]. In contrast to this, Moya-Perez et al. [177] showed no change in diversity. MS has been shown to have the ability to increase systemic inflammation, and systemic microbial loads through gut barrier dysfunction [176,177]. Levels of toxic bacteria have also been found to be upregulated in mice [176], and symptoms of IBS (intestinal hyperpermeability, visceral hypersensitivity, microbial dysbiosis, and low-grade intestinal inflammation) were found as a result [178]. Gareau et al. [179] showed increased adhesion and penetration of total bacteria in the gut, along with significantly reduced levels of *Lactobacillus* species. Notably, levels of bacteria and SCFA-producing genera can be dysregulated due to MS. Significantly higher levels of SCFA-producing genera such as *Fusobacterium* and *Clostridium* have been found in the gut, and this was shown to be positively correlated with the degree of visceral hypersensitivity [180,181].

Physical stressors are another popular form of stress paradigm used in animal studies. In a study on germ-free mice, it was demonstrated that a stress paradigm induced an exacerbated release of corticosterone and adrenocorticotropic hormone when compared to controls [171]. In addition to this, Table 3 shows the effects on the gut microbiome of various RIS studies in various mouse and rat models. Alterations of gut microbiota including increased kynurenine toxicity, changes in bacterial makeup, downregulation of microbiome flora richness, and disrupted alpha and beta indices of diversity have been observed [182,183,184,185,186,187]. As seen in MS stress-induced studies, intestinal hyperpermeability was observed due to RIS and tight junction protein levels (ZO-1, Occludin, and Claudin1) were disrupted [188,189]. Levels of *Akkermansia*, a common mucin-degrading bacteria, were significantly decreased in stressed animals compared to controls. This bacteria is inversely correlated with inflammation, diabetes, and obesity [190]. Together, studies revealing stress-related gut dysbiosis provide support for the critical role the microbiota may have in the development of the stress response and the effect that stress has on neurodegenerative diseases, such as AD.

Interestingly, the HPA axis, the main stress axis in mammals, has been shown to be in close communication with the gut microbiome, and one of the important methods of communication between the gut and the brain [191]. Activation of the HPA axis causes a release of CRF from the hypothalamus, and CRF becomes one of the main stress-related neuropeptides involved with both the brain and gut [191]. Stress-induced activation of the HPA axis by MS increases levels of corticosterone, CRF, and has been shown to also affect the permeability of the intestinal wall [192,193,194]. Both CRF and the family of neuropeptides urocortin (Ucns) are shown to have proinflammatory and anti-inflammatory effects on the GI. Regulating the release of these peptides alters the inflammation in the GI tract and plays a role in increasing intestinal barrier leakage [193,195,196].

**Table 3 biomedicines-11-01884-t003:** Studies of stress effects on gut microbiome from common methods: maternal separation and restraint stress.

Stressor	Animal Model	Findings	Reference
Maternal Separation	C57BL/6J (male)	Early life stress-induced gut microbiota alterations and lasting CNS inflammation Rescued by probiotics Probiotics improve glucocorticoid sensitivity	[177]
	C57BL/6J (male)	Significant reduction of serum TNF-a and increase in IL-6 Reduction of IL-6 levels after probiotic ingestion	[197]
	C57BL/6J (male)	Differential abundance of gut microbiota in maternal separation groups differed from control groups Stress exposure reduced the alpha diversity and altered microbial community at PD28	[198]
	C57BL/6J (male)	Maternal separation combined with chronic unpredictable stress paradigms increased systemic microbial load through gut barrier dysfunctions Stress-related increases in Clostridium were observed	[176]
	C57BL/6N (both sexes)	Maternal separation-induced changes in animals lead to intestinal dysbiosis	[199]
	NMRI mice (both sexes)	Maternal separation altered the composition of gut microbiota	[200]
	C3H/HeN mice (male)	Maternal separation induces microbiota dysbiosis in favor of pathobionts IL-17 and IL-22 decreased in response to glucose intolerance in stressed animals compared to controls	[201]
	C3H/HeN mice (both sexes)	Maternal separation induced the main features of IBS (intestinal hyperpermeability, visceral hypersensitivity, microbiota dysbiosis, and low-grade intestinal inflammation	[178]
	SD (male)	Maternal separation reduced swim behaviour and decreased mobility in forced swim test Stress-related decrease in brain noradrenaline and increase in peripheral IL-6 Probiotic ingestion normalized these levels	[202]
Maternal Separation	SD (both sexes)	Maternal separation increased adhesion/penetration of total bacteria in gut and significantly reduced *Lactobacillus* species Stress-related elevation of corticosterone levels observed Probiotic ingestion ameliorated gut penetration and restored corticosterone levels	[179]
	SD (both sexes)	Maternal separation pups showed an adult-like profile of long-lasting fear memories and fear relapse following extinction Probiotic-treated pups exhibited age-appropriate infantile amnesia and resistance to relapse	[203]
	SD (both sexes)	Maternal separation disrupted the Firmicutes-to-Bacteroides ratio in the gut Stress-related disruption of acetate, propionate, and butyrate in fecal samples was observed This was restored by a probiotic mixture	[204]
	SD (both sexes)	Significant differences in microbial community of gut in Maternal separation group compared to control	[205]
	SD (both sexes)	Maternal separation showed a difference in abundance of various bacteroids in fecal samples of both sexes Levels of proinflammatory cytokines were increased in the colon and sera of male stressed rats	[206]
	SD (both sexes)	Maternal separation animals showed significantly lower levels of SCFA producers	[180]
	SD (both sexes)	Maternal separation animals showed a difference in SCFA-producing genera, Fusobacterium and Clostridium compared to controls Fusobacterium stress-related increase positively correlated with the degree of visceral hypersensitivity	[181]
	SD (both sexes)	Maternal separation triggered gut microbiota composition changes compared to control groups Gut dysbiosis was reversed by probiotic ingestion (*Lactobacillus*)	[142]
Maternal Separation	SD (both sexes)	Maternal separation produced alterations in the structure and composition of the gut microbiota Significant differences in bacterial types in stressed animal microbiomes compared to controls	[207]
	SD (both sexes)	Maternal separation rats showed hypercorticosteronemia, enhanced intestinal permeability, and changes in gut microbiota structure Probiotic feeding prevented these changes	[208]
	Wistar rats (male)	Significant gut barrier dysfunction in maternal separation groups compared to control ML-7 (MLCK inhibitor) strengthened the intestinal barrier and restored levels of numerous bacteria in the gut	[173]
Physical Restraint	C57BL/6J (male)	Chronic restraint stress exacerbated kynurenine (Kyn) toxic signaling in the gut, especially the colon Indoleamine 2,3-dioxygenase was upregulated in the brain and gut, promoting transfer of Tryptophan metabolic pathway to Kyn signaling Stressed mice showed enhanced intestinal permeability compared to controls	[182]
	C57BL/6J (male)	Restraint stress increased intestinal hyperpermeability and disrupted tight junction proteins (ZO-1, Occludin, and Claudin1) Increased inflammation in enteric glial cells increased relative abundance of fecal *Akkermansia* (mucin-degrading Gram-negative bacteria)	[188]
	C57BL/6J (male)	Chronic restraint animals showed decreased levels of SCFA in feces compared to controls Stressed animals showed gut dysbiosis, and decreased levels of Occludin and Claudin-1, correlating with decreased intestinal barrier function	[189]
	C57BL/6J (both sexes)	Chronic restraint stress revealed sex differences in fecal microbiota makeup Fecal transplant of stressed mice to germ-free mice decreased pain threshold, and resulted in further sex differences in gut-microbiota makeup	[209]
Physical Restraint	C57BL/6N (male)	Chronic restraint mice exhibited alterations in microbiota composition, disruption of colonic mucus, and aggravation of colitis Abundance of *Akkermansia* was significantly decreased in stressed mice compared to controls	[183]
	C57BL/6 specific-pathogen free (both sexes)	*Lactobaccillus* animalis was enriched in stress group, and positively correlated to behavioural deficiencies	[210]
	ICR mice (male)	Microbiome flora richness was significantly lower in the stress group compared to controls	[184]
	Wistar rats (male)	Chronic restrain stress increased gut dysbiosis compared to controls Both alpha and beta diversity indexes were increased in stress animals	[185]
	SD (male)	Chronic restraint stress enhanced the abundance of bacteroids and altered gut microbiota and metabolites Altered gut microbiota was correlated with PI3K/Akt/mTOR pathways Chronic restraint stress decreased phosphorylation of PI3K/Akt/mTOR pathway in microglia and enhanced LPS-induced microglia activation	[211]
	SD (male)	Restraint-stressed rats exhibited dysregulated gut microbiota Fecal B-d-glucosidase activity differed from control rats, leading to systemic exposure to ginsenoside RB1 and its metabolites	[186]
	SD (both sexes)	Chronic restraint stress-induced dysbiosis	[187]

#### 3.3.2. Role of Stress-Induced Gut Dysbiosis in AD

With stress having a lasting impact on gut health and contributing to gut dysbiosis, many physiological and behavioural consequences have also been highlighted. Stress-induced increased inflammation and impaired cognition have been found in animal models of various neurodegenerative diseases, depression, and irritable bowel syndrome (IBS) [212,213]. Specifically, the intestinal barrier has been found to be damaged in AD patients and animal models [214,215]. Gut microbes influenced by stress exposure can impact intestinal barrier function and ultimately lead to intestinal permeability [182,188,203,216]. Interestingly, when probiotic *Lactobacillus* was administered in rodents, this barrier leakiness was prevented, along with improved behavioural, cognitive, and biochemical parameters [213,217].

A correlation between increased dysbiosis and stress in the gut was shown to be related to the PI3K/Akt/mTOR pathways [185]. Chronic stress decreased phosphorylation of this pathway in microglia and enhanced LPS-induced microglial activation [185]. This is consistent with AD research, in that the activation of this pathway subsequently produces an increase in GSK-3B-induced tau phosphorylation [218].

Alterations in the gut–brain axis and the composition of gut microbial species are seen in aging. LPS, neuroinflammatory cytokines, and amyloids, which may contribute to the pathogenesis of AD, have been hypothesized to pass through the compromised gastrointestinal tract (GI) tract and BBB [219,220,221]. The increased leakiness of the intestinal barrier from gut dysbiosis can increase the abundance of toxic movement from the gut to the brain, and enhance disease progression. Stress paradigms such as MS have been associated with increased toll-like receptor 3,4 and 5 mRNA expression [222]. Implications of this increase can include higher susceptibility to infection, higher levels of pathogenic bacteria in the gut, and an increase in cytokine production, which can seed neuroinflammation in AD [222]. In addition to this, downregulation of levels of the gene expression encoding the glucocorticoid receptor, which is responsible for down-regulating inflammation in the cortisol response, was found in models of early life stress [118]. In the same study, it was also found that exposure to stress at a young age increased the expression of toll-like receptor (TLR) 4, activating the innate immune system response, and increasing inflammation [118].

## 4. Therapeutic Potential of Probiotics on Gut Dysbiosis

Probiotics have been shown to maintain human intestinal health and are defined as live microorganisms which confer a health benefit on the host by the World Health Organization [223]. Alteration of the gut microbiota can be achieved through the supplementation of probiotics, although the viability when administered remains questionable in some cases. Two commonly used probiotic species, *Lactobacillus* and *Bifidobacterium*, have been shown to have promising effects on host health by changing the gut microbiota composition. In healthy subjects, probiotic supplementation of these reveals a decrease in anxiety, depression, and stress-related behaviors [224,225]. Both of these species have been shown to inhibit harmful bacteria, improve gastrointestinal barrier function, and suppress proinflammatory cytokines [226]. Certain probiotics have the potential of suppressing AD-related GSK3B overexpression, by modulating the PI3K/Akt signaling pathways in the gut or indirectly through metabolites and SCFAs to provide changes in the gut and brain [227,228,229]. Multiple probiotic strains have been shown to promote the survival and growth of existing neurons and have positive effects on learning, memory, and cognition through increased BDNF levels, neuropeptides, or neurotransmitters in animal models [230,231,232,233].

Further strengthening the role of the gut microbiota in the stress response, probiotic supplement feeding has shown promising results for restoring and normalizing levels of gut microbiota bacteria and strengthening gut barrier function. Studies highlighting the ability of probiotics to recuse stress-related deficiencies in behavioural and biological parameters are reviewed in Cryan et al. [187] and shown in Figure 1. As seen in Table 3, probiotics can rescue deficits in the gut caused by both MS and RIS. Gut microbiota alterations and dysbiosis, increased adhesion, permeability, and penetration of the gut barrier, and the disrupted Firmicutes-to-Bacteroides ratio in the gut were all rescued in models supplemented with *Lactobacillus* [177,179,204,208]. Levels of peripheral neuroinflammatory cytokines, notably IL-6, which have been observed to be increased in stress animals, can be normalized through probiotic ingestion [197,202]. Neuroinflammation both in the CNS and PNS is improved after feeding [177].

Probiotic supplementation has also been shown to improve AD pathologies, as reviewed by Peterson et al. [234]. Multiple studies on AD patients show promising therapeutic effects on AD progression, including improved learning and memory, decreased oxidative stress, modified insulin resistance, improved behavioural performance on visual-spatial and executive/language tasks, decreased inflammation, mitochondrial dysfunction, gut barrier permeability, and DNA damage [234,235]. These effects are repeated and in some cases strengthened when combined with herbs or selenium treatment [236,237].

## 5. Conclusions

Taken together, stress paradigms in animal studies are shown to affect AD development and gut microbiota makeup, causing dysbiosis. The order in which these events occur is unclear; however, the relationship between the two cannot be overlooked. Further strengthening the role of the gut microbiota in the stress response, probiotic supplement feeding has shown promising results for restoring and normalizing levels of gut microbiota bacteria and strengthening gut barrier function. Combined with this, studies using probiotic supplementation have also been shown to improve AD pathologies [233]. Additionally, enrichment may be useful as a complement to other treatment methods in AD, helping to reduce cognitive symptoms and possibly slow pathological progression and potentially reverse stress-induced gut dysbiosis, which in turn could lead to further improvement. Given that AD is often comorbid with other neuropsychiatric disorders, including depression, bipolar disorder, and schizophrenia as outlined by Garcez and colleagues [238], enrichment may also help better those comorbidities or lessen their severity alongside AD. Combining enrichment (cognitive and exercise) and gut health can provide an optimal lifestyle that can serve as a beneficial strategy to combat stress and AD.

Additional studies using 16sRNA sequencing are needed in this field to establish a clear picture of which gut bacteria have a direct effect on AD development and the order in which these changes occur. Additionally, higher importance must be placed on sex differences in gut microbiota stress and enrichment studies. Overall, the understanding of the roles enrichment, stress, and gut dysbiosis has in AD development provides a possibility for non-invasive therapeutic potential through both enrichment and probiotic supplementation. Behaviorual intervention and gut microbiota manipulation could provide exciting new avenues to slow, stop, or prevent AD progression in both pre-clinical and prodromal stages.

## Figures and Tables

**Figure 1 biomedicines-11-01884-f001:**
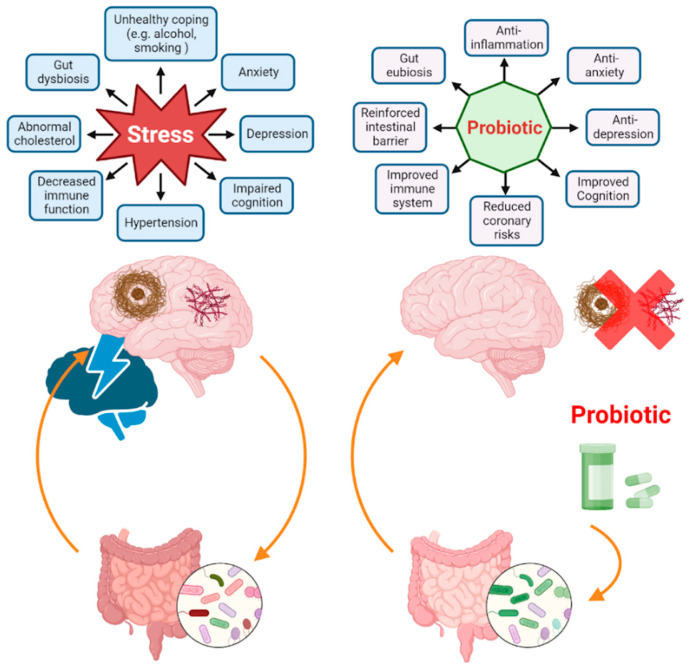
Effects of stress and probiotic manipulation on the gut microbiota and health.

**Table 1 biomedicines-11-01884-t001:** Studies examining the effect of enrichment on Alzheimer’s disease development and progression.

Enrichment Paradigm	Animal Model	Findings	Reference
Cognitive Enrichment	3xTg-AD (both sexes—mouse)	Repeated spatial learning training reduced cognitive decline in a separate spatial task	[64]
Environmental Enrichment	E257K/P301S-Tau-Tg (both sexes—mice)	Increased neurogenesis Improved cognition Reduced tau pathology	[59]
	apoE4 (male—mouse)	Improved neurogenesis in apoE3 controls Worsened in apoE4 animals	[108]
	APP^swe^/PS1^ΔE9^ (female—mice)	Increased Aβ burden	[109]
	APP^Sw,Ind^ (female—mouse)	Improved cognitive performance Increased neurogenesis	[63]
	APP^swe^/PS1^ΔE9^ (male—mice)	Increased neurogenesis Reduced Aβ and ptau	[69]
Environmental Enrichment	TgCRND8 (female—mouse)	Increased neurogenesis Reduced Aβ burden	[70]
	Aβ1-42 infusion at CA1 (male—rat)	Improved neurogenesis in Aβ animals	[71]
	5xFAD (male—mouse)	Reversed cell death and reduced neurogenesis of Aβ seeded animals	[72]
	Tg4-42^hom^ (both sexes—mouse)	Preserved spatial and recognition memory	[74]
	AD11 (both sexes—mouse)	Improved memory and reduced Aβ burden	[75]
	oAβ intraventricular injection (male—mouse)	Prevented Aβ-mediated changes in microglia and inflammatory gene mRNA	[76]
	PDBFG-APP^Swlnd^ (both sexes—mouse)	Improved spatial memory Reduced Aβ burden	[77]
	Tg2576 (female—mouse)	Reversed parvalbumin deficit	[78]
	SAMP8 (male—mouse)	Increased spatial memory Increased synaptic plasticity Reduced apoptosis Reduced Aβ burden	[79]
	Chronic cerebral hypoperfusion (male—rat)	Improved spatial memory Reduced inflammation	[80]
	5xFAD (female—mouse)	Increased exploration Reduced cognitive deficits Reduced AD pathological markers	[81]
	APP^swe^/PS1^ΔE9^ (male—mouse)	Reduced Aβ deposition Increased Aβ-degrading enzymes	[82]
	PS1/PDAPP (both sexes—mouse)	Increased cognitive performance Reduced Aβ burden	[83]
	APP23 (male—mouse)	Improved spatial memory at 7–8 mo. Reduced Aβ deposition by 18 mo.	[88]
	High sucrose or high-fat diet (male—rat)	High sucrose and fat diets increased AD pathology Enrichment returned to control levels	[84]
	APP^Swe,Ind^, high-fat diet (both sexes—mouse)	High-fat diet worsened memory and pathology Ameliorated by enrichment	[85]
Environmental Enrichment	Tg2576 (female—mouse)	Protective against cognitive decline and Aβ but did not rescue later	[89]
	3xTg-AD (both sexes—mouse)	Improved immune system activity for males	[87]
	APP^swe^/PS1^ΔE9^ (male—mouse)	Improved synaptic health but not cognitive abilities	[95]
	5xFAD (male—mouse)	Prevented learning impairment Increased synaptic plasticity Reduced inflammation	[96]
	3xTg-AD (female—mouse)	Reduced anxiety-like behaviour	[97]
	TgCRND8 (female—mouse)	Improved exploratory behaviour Reduced anxiety-like behaviour No memory improvement	[98]
	5xFAD (female—mouse)	Increased survival rate Improved motor skills No improvement of anxiety, Aβ, or inflammation	[99]
	APP^swe^/PS1^ΔE9^ (male—mouse)	Improved short-term spatial memory Prevented increases in inflammation	[100]
	APP/PS1KI (female—mouse)	Mild improvement in motor skills No changes to memory, Aβ, or neuronal loss	[101]
	SHR72 (male—rat)	Improved motor skills Mildly improved inflammation Did not influence ptau	[102]
	APP^sw^ (both sexes—mouse)	Increased spatial memory despite continued Aβ deposition	[103]
	PDAPP-J20 (female—mouse)	Reduced Aβ Improved astroglial response toward plaques	[104]
	3xTg-AD (both sexes—mouse)	Mild improvements to spatial learning in advanced stages	[105]
	SHR72 (male—rat)	Increased spatial memory performance Reduced ptau in mild cases	[106]
	APP^swe^/PS1^ΔE9^ (female—mouse)	Improved spatial memory Reduced Aβ deposition	[107]
	APP^Swe^/PS1^L166P^ (female—mouse)	Reduced spatial memory impairment Worsened Aβ deposition	[110]
	APP^swe^/PS1^ΔE9^ (male—mouse)	Improved spatial memory Reduced Aβ in motor and sensory cortices	[111]
Physical Enrichment	APP^swe^/PS1^ΔE9^ (male—mouse)	Improved cognition Reduced inflammation	[65]
Environmental and Physical Enrichment	APP23 (female—mouse)	At 18 mo. time point, both enrichment types reduced Aβ and increased neurogenesis	[66]
	3xTg-AD (both sexes—mouse)	EE alone reduced inflammation Combined EE and PE reduced to control levels	[67]
	3xTg-AD (both sexes—mouse)	Both enrichment types reversed deficits in neurogenesis	[61]
Environmental, Physical, and Social Enrichment	Aβ infusion at CA1 (male—rat)	EE and PE prevent memory impairment SE prevents social recognition impairment	[60]
	APP/PS1 (both sexes—mouse)	EE alone prevented cognitive impairment, reduced Aβ, and increased synaptic plasticity	[86]
Environmental Enrichment with *B. breve*	Aβ1-42 infusion at hippocampus (male—mouse)	Combined treatment rescues impaired cognitive performance	[113]
Environmental Enrichment with Magnesium	APP^swe^/PS1^ΔE9^ (female–mouse)	EE alone improved short-term memory Combined treatment with MgT improved long-term memory	[112]
Environmental Enrichment with Resveratrol	AlCl3 (male—mouse)	Both treatments individually reduced Aβ Combined treatment worsened	[117]
Environmental Enrichment with Caffeine	Tg4-42 (both sexes—mouse)	Combined treatment improved spatial memory, but did not add to individual effects of EE or Caf alone on motor skills, recognition, or neurogenesis	[116]
Environmental Enrichment with Donepezil	Aβ1-42 infusion at hippocampus (male—rat)	Combined treatment only was able to restore spatial memory and elevate BDNF to control levels	[115]
Environmental Enrichment with Memantine	SAMP8 (male—mouse)	Both treatment options effective, but more so when combined, at improving spatial memory and reducing ptau and Aβ	[114]
Gestational Enrichment	3xTg-AD (both sexes—mouse)	Offspring of enriched dams had preserved synaptic plasticity and memory	[118]
	5xFAD (female—mouse)	While pregnancy and lactation worsen Aβ and neurogenesis, EE rescues to non-pregnant levels	[90]
Neonatal Handling	3xTg-AD (both sexes—mouse)	Handling provided mild protective effects against spatial memory decline, especially in females	[93]
	3xTg-AD (both sexes—mouse)	Handling improved performance in forced swim test	[92]
Environmental Enrichment and Maternal Separation	htauE14 infusion at locus coeruleus (both sexes—rat)	EE reduced anxiety-like behaviour Both types of enrichment improved spatial recall	[62]
Environmental Enrichment and Stress	Tg2576 (female—mouse)	EE counteracted spatial memory deficits induced by stress, reduced ptau, and improved neurogenesis	[73]
Olfactory Enrichment	WT (male—rat)	Long-term, but not short-term enrichment reduced ptau	[94]
Environmental and Social Enrichment	Tg-SwDI (female—mouse)	SE, but especially EE, improved motor skills and cognitive performance	[68]
Environmental Enrichment in Social Isolation	APP^swe^/PS1^ΔE9^ (male—mouse)	EE reversed cognitive decline seen in isolation, reduced apoptosis and inflammation	[58]

**Table 2 biomedicines-11-01884-t002:** Studies examining the effect of stress on Alzheimer’s disease development and progression.

Stress Paradigm	Animal Model	Findings	Reference
Chronic Unpredictable Stress	APP^swe^/PS1^ΔE9^ (male—mouse)	Increased Aβ deposition	[122]
	Aβ infusion in lateral ventricle (male—rat)	Increased ptau accumulation	[18]
	WT (male—rat)	Induced ptau, especially with repeat exposure, in hippocampus and PFC	[134]
	Tg2576 (female—mouse)	Worsened spatial memory Increased ptau and Aβ	[136]
	Tg2576 (female—mouse)	Accelerated spatial memory decline Increased ptau and Aβ	[137]
	SAMP8 (male—mouse)	Worsens spatial memory deficit Reduces synaptic plasticity	[138]
	arcAβ (male—mouse)	In WT controls, stress impaired attention and impulsivity In Aβ mice, stress reduced impulsivity	[139]
	APP^swe^/PS1^ΔE9^ (male—mouse)	Increased Aβ deposition Increased glucocorticoids	[140]
	apoE4-TR (male—mouse)	apoE4 mice were more susceptible to stress-induced cognitive decline and depression than controls	[141]
Chronic Unpredictable Stress with Ace Inhibitor	WT (male—rat)	Stress-induced ptau development, mediated through ACE enzyme activity	[129]
Chronic Unpredictable Stress with Escitalopram	WT (male—rat)	Stress increased ptau burden Escitalopram reduced ptau	[133]
Chronic Unpredictable Stress and Maternal Separation	htauE14 infusion at locus coeruleus (both sexes—rat)	CUS worsened, while maternal separation improved, discrimination learning in htauE14 animals	[62]
Maternal Separation	APP^swe^/PS1^ΔE9^ (male—mouse)	Reduced spatial memory Increased Aβ deposition	[165]
	APP^NL-G-F^ (male—mouse)	Induced Aβ deposition Impaired spatial memory Increased inflammation	[166]
Model Mimics Stress Response	corticotropin-releasing factor overexpression (female—mouse)	CRF-OE increases ptau Treatment with CRFR-1 antagonist reduces phosphorylation	[119]
Dexamethasone Administration	Tg2576 (female—mouse)	Impaired fear recall Increased Aβ	[123]
Isolation	Tg2576 (both sexes-mouse)	Increased corticosterone and CRFR-1 expression Increased Aβ deposition	[127]
Isolation and Environmental Enrichment	5xFAD (male—mouse)	Increased Aβ Reduced fear learning EE did not rescue those effects	[130]
Isolation and Restraint	Tg2576 (both sexes—mouse)	Long-term isolation and short-term restraint increased Aβ, mediated through CRF	[158]
Psychosocial Stress	Aβ1-42 infusion at left ventricle (male—rat)	Stress worsened spatial memory	[126]
Gestational Restraint	APP^swe^/PS1^ΔE9^ (both sexes—mouse)	Males had reduced spatial memory and HPA response Females had increased spatial memory and depressive-like symptoms and reduced Aβ	[155]
Restraint	TgF344-AD (male—rat)	No significant changes in anxiety- or fear-like behaviour	[164]
	TgCRND8 (female—mouse)	Restraint did not worsen Aβ pathology	[163]
	APP-CT100, Tg2576 (both sexes—mouse)	Accelerated cognitive decline Increased Aβ and ptau pathology	[161]
	APP^swe^/PS1^ΔE9^ (male—mouse)	Increased inflammation	[159]
	L/V-Tg (male—mouse)	Short-term stress-induced neurodegeneration Long-term also reduced neurogenesis	[157]
	5xFAD (both sexes—mouse)	Increased Aβ deposition in females	[121]
	APP/hAβ/PS1 (male—mouse)	AD model animals more susceptible to effects of stress Stress exacerbated AD pathology	[124]
Chronic Unpredictable Stress and Restraint	Tg2576 (female), P301S (male), CRF-OE (female) (mouse)	Restraint increased Aβ and ptau pathology and worsened memory in AD models, not in CRF-OE	[120]
Restraint with *ALOX5* Knockout	3xTg-AD (female—mouse)	Stress worsened ptau and impaired fear memory but not in *ALOX5* KO animals	[123]
Restraint with Icariin	APP/PS1 (male—mouse)	Stress-induced depressive-like behaviour, worsened spatial memory, and increased Aβ. Icariin reversed memory impairment and reduced Aβ	[156]
Restraint with Memantine	WT (male—mouse)	Restraint-induced ptau by day 16 MEM dose-dependently reduced ptau On day 28 MEM worsened ptau	[91]
Restraint with CRFR NBI 27914	Tg2576 (both sexes—mouse)	Restraint increased Aβ and ptau pathology NBI prevented such effects	[128]
Restraint with PNU-282987	APP^swe^/PS1^ΔE9^ (male—mouse)	Stress-impaired spatial memory PNU did not rescue	[162]
Sleep Deprivation	WT (male—rat)	Reduced spatial learning Increased Aβ	[131]
	WT (male—mouse)	Reduced fear learning Increased Aβ	[130]
	APP^swe^/PS1^ΔE9^ (male—mouse)	Reduced spatial memory Increased Aβ and ptau	[154]
	APP^swe^/PS1^ΔE9^ (female—mouse)	Fragmented sleep increased Aβ Severity of sleep deprivation correlated with amount of Aβ	[153]
	APP^swe^/PS1^ΔE9^ (both sexes—mouse)	Impaired spatial memory Increased Aβ and ptau	[152]
	5xFAD (both sexes—mouse)	AD model animals had more fragmented sleep and were more susceptible to effects of sleep deprivation	[143]
	3xTg-AD (both sexes—mouse)	Poorer spatial memory Reduced synaptic plasticity	[149]
	P301S (male—mouse)	AD model mice had reduced sleep which worsened with age Reduced sleep correlated with increased ptau	[147]
	APP^swe^/PS1^ΔE9^ (male—mouse)	Reduced spatial and fear memory Reduced synaptic plasticity Increased Aβ Increased inflammation	[146]
Sleep Deprivation	APP/PS1 (both sexes—mouse)	Poorer spatial memory Increased inflammation Reduced synaptic plasticity Increased Aβ	[145]
	Aβ1-42 infusion at lateral ventricle (male—mouse)	Aβ infusion disrupted sleep patterns Sleep-deprived animals more prone to Aβ effects on memory	[144]
	3xTg-AD (male—mouse)	Increased corticosterone Reduced spatial memory Increased Aβ and ptau	[125]
Sleep Deprivation and Acoustic Stimulation	3xTg-AD (both sexes—mouse)	Sleep deprivation worsened spatial memory and increased Aβ and ptau Acoustic stimulation reversed such effects	[148]
Sleep Deprivation and Ketogenic Diet	WT (female—mouse)	Sleep deprivation worsened spatial memory and increased Aβ and ptau Keto diet reversed such effects	[135]
Sleep Deprivation and Orexin Receptor Antagonist	Tg2576 (sex not specified—mouse)	Sleep deprivation increased Aβ deposition Aβ reduced following almorexant treatment	[151]
Sleep Deprivation and Orexin Knockout	APP^swe^/PS1^ΔE9^ (both sexes—mouse)	Orexin KO mice had reduced Aβ and increased sleep time Rescuing orexin neurons increased Aβ	[150]

## Data Availability

Not applicable.

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
