# Peer review of "Life Experience Matters: Enrichment and Stress Can Influence the Likelihood of Developing Alzheimer’s Disease via Gut Microbiome"

_biomedicines, 2023, doi:10.3390/biomedicines11071884_

Round 1

Reviewer 1 Report

The article presented to me for review, "Life Experience Matters: Enrichment and Stress can Influence the Likelihood of Developing Alzheimer's Disease via Gut Microbiome," treats the medically relevant topic of Alzheimer's disease. 

The authors have discussed the topic of stress, the gut microbiota and the relationship of these factors to Alzheimer's disease in an extremely precise and detailed way.

In Chapter 3, they also discussed animal studies related to the topic of the paper.

They included 3 tables and 1 figure in their paper.

In my opinion, the authors drew appropriate and relevant conclusions from their work.

It should be noted that the authors cited 245 current literature items in their work.

In conclusion in my opinion the work is very interesting, deals with an important medical topic and can be accepted for publication in its present form.

Reviewer 2 Report

Alzheimer's dementia exacts a cruel personal and familial toll and an expensive burden on societies. Attempts at therapies using drugs (20th century) and antibodies to remove aggregated proteins that may be toxic to neurons (21st century) have yielded minimal results. There are compelling needs for interventions that improve cognition and potentially reverse deficits in AD subjects.

Recently there is increased investigation of the gut microbiome's influence on brain function and brain diseases. The current paper is an excellent review of the relationships among stress, Alzheimer's dementia and gut microbiome alterations. The authors are to be congratulated on comprehensively reviewing all three components and integrating the resulting information into sound recommendations.

The one revision suggestion from me is to modify the last section dealing with probiotic treatments. As the authors well know, "probiotic" is a non-specific term that does not define bacterial content or viability when administered. The final section ends up detracting from an otherwise excellent article. I suggest reducing the length of the last section and refer to the referenced probiotic studies in a more general manner.

English language is excellent. There are a very few minor errors in language that are easily corrected at the editing stage.
